Prevention of necrotizing enterocolitis with probiotics: a systematic review and meta-analysis

Sawh Sonja C. sonja.sawh@lhsc.on.ca 1
Deshpande Santosh 1
Jansen Sandy 1
Reynaert Christopher J. 1
Jones Philip M. 2
1 Department of Pharmacy, London Health Sciences Centre , London , Ontario , Canada
2 Departments of Anesthesia & Perioperative Medicine and Epidemiology & Biostatistics, University of Western Ontario , London , Ontario , Canada
Kuhle Stefan
Electronic publication date: 2016 Oct 5
Publication date: 2016
Volume: 4
Electronic Location ID: e2429
Received 2016 Mar 16; Accepted 2016 Aug 10
Copyright: ©2016 Sawh et al.
Copyright year: 2016
Copyright holder: Sawh et al.
License: This is an open access article distributed under the terms of the Creative Commons Attribution License, which permits unrestricted use, distribution, reproduction and adaptation in any medium and for any purpose provided that it is properly attributed. For attribution, the original author(s), title, publication source (PeerJ) and either DOI or URL of the article must be cited.
License URL: https://creativecommons.org/licenses/by/4.0/

Keywords: Probiotics, Necrotizing enterocolitis, Lactobacillus, Bifidobacterium, Saccharomyces, Extremely premature, Newborn, Infant, Premature, Enterocolitis

Funding: Pharmacy Department, London Health Sciences Centre This research was supported by the Pharmacy Department, London Health Sciences Centre. The funders had no role in study design, data collection and analysis, decision to publish, or preparation of the manuscript.

==============================
Context

Necrotizing enterocolitis (NEC) is the most frequent gastrointestinal emergency in neonates. The microbiome of the preterm gut may regulate the integrity of the intestinal mucosa. Probiotics may positively contribute to mucosal integrity, potentially reducing the risk of NEC in neonates.

Objective

To perform an updated systematic review and meta-analysis on the efficacy and safety of probiotics for the prevention of NEC in premature infants.

Data Sources

Structured searches were performed in: Medline, Embase, and the Cochrane Central Register of Controlled Trials (all via Ovid, from 2013 to January 2015). Clinical trial registries and electronically available conference materials were also searched. An updated search was conducted June 3, 2016.

Study Selection

Randomized trials including infants less than 37 weeks gestational age or less than 2,500 g on probiotic vs. standard therapy.

Data Extraction

Data extraction of the newly-identified trials with a double check of the previously-identified trials was performed using a standardized data collection tool.

Results

Thirteen additional trials (n = 5,033) were found. The incidence of severe NEC (RR 0.53 95% CI [0.42–0.66]) and all-cause mortality (RR 0.79 95% CI [0.68–0.93]) were reduced. No difference was shown in culture-proven sepsis RR 0.88 95% CI [0.77–1.00].

Limitations

Heterogeneity of organisms and dosing regimens studied prevent a species-specific treatment recommendation from being made.

Conclusions

Preterm infants benefit from probiotics to prevent severe NEC and death.

Introduction

Rationale

Necrotizing enterocolitis (NEC) is a gastrointestinal (GI) syndrome characterized by transmural inflammation and necrosis of the large or small bowel and subsequent translocation of gas-forming organisms into the intestinal wall (Thompson & Bizzarro, 2008; Morgan, Young & McGuire, 2011). Primarily seen in infants, the incidence of NEC is inversely correlated with gestational age (GA) and birth weight (Sharma & Hudak, 2013; Choi, 2014). The overall incidence of NEC in all infants ≤33 weeks GA in a survey of Canadian neonatal intensive care units (NICUs) was 5%, and 7% for infants less than 1500 g birth weight in 2013 (Henry & Moss, 2008).

The consequences of NEC are potentially devastating—20% to 40% of patients require surgical intervention and mortality ranges from 15% to 30% (Holman et al., 2006; Sia et al., 2014). Survivors of NEC risk significant morbidity including short gut syndrome, strictures, and neurodevelopmental impairment (Henry & Moss, 2008; Holman et al., 2006).

The signs and symptoms of NEC were classified by Bell in 1978 and gave rise to modified criteria for diagnosis of NEC in 1996 by Neu (Sia et al., 2014). The modified Bell’s criteria describe the systemic clinical signs of NEC, the important GI signs (which can help differentiate NEC from sepsis), and the radiologic features.

The immature GI tract of preterm infants is particularly susceptible to mucosal injury from a variety of factors. Intestinal and immunological deficiencies associated with prematurity, enteral feeding, microbial overgrowth, and circulatory instability have all been implicated in the pathogenesis of NEC (Schanler, 2015).

Recent research has focused on microbial overgrowth in the GI tract of premature infants, with an overabundance of pathogenic organisms and lack of microbial diversity being key discoveries. These observations imply that a disturbance in the microbiome, and not a single pathogen, may be a causative factor of NEC (Terrin, Scipione & De Curtis, 2014). The lower prevalence of protective Lactobacillus or Bifidobacterium species in preterm infants compared to term infants make probiotics a potential intervention for the prevention of NEC (Morowitz et al., 2010).

Previous systematic reviews

At the time of our search, there were two recent systematic reviews and meta-analyses on this topic (AlFaleh, 2014; Yang et al., 2014). The Cochrane review on this topic is thorough, but it was last updated in October 2013 (AlFaleh, 2014). The Yang (Terrin, Scipione & De Curtis, 2014) review included many of the same studies but included additional studies as a result of a Chinese trial database search (Ke, Su & Li, 2008; Huang, Yang & Huang, 2009; Ren, 2010; Di & Li, 2010). Since the publication of these two systematic reviews, more large randomized clinical trials have been published.

Objective

The objective of this systematic review was to assess the efficacy and safety of probiotics for the prevention of NEC in premature infants. We planned to update the previous systematic reviews (AlFaleh, 2014; Yang et al., 2014) using similar eligibility criteria.

Methods

Protocol/registration

The systematic review methods and analysis plans were undertaken according to published guidelines by PRISMA (Moher et al., 2009).

Eligibility criteria

Studies: All randomized clinical trials were considered for inclusion. No language restrictions were applied.

Participants: Infants of less than 37 weeks gestation or weighing less than 2,500 g at birth.

Interventions: Probiotics in any species and any dose, or prebiotic/probiotic combinations (synbiotics) of any species and any dose.

Comparators: Probiotic products with different species than the intervention group (i.e., RCTs comparing one species to another head-to-head), placebo, or standard therapy.

Outcomes: The primary outcome of the review was the incidence of severe NEC (Bell’s Stage 2 or greater). Secondary outcomes included all-cause mortality, all-cause sepsis, culture-proven sepsis, bacterial sepsis, fungal sepsis, length of stay in hospital, time to achieve full feeds, duration of parenteral nutrition, and weight gain.

Outcome definitions

1. Sepsis was accepted as defined by the authors of the individual trials.

2. Culture-proven sepsis was accepted as defined by the authors but needed to include a positive culture (blood, urine, or cerebrospinal fluid) to qualify.

3. Length of stay in hospital and length of stay in NICU were considered equivalent. Many studies discharged infants home directly from the NICU.

4. We considered the outcome “age at which full enteral feeding was reached” to be the same as “time to reach full feeds.” We considered the “age at which parenteral nutrition stopped” to be the same as the outcome of “duration of parenteral nutrition”.

5. We subgrouped trials in duration of therapy categories based on the durations reported in the results section of each paper, not the planned duration. We only placed a trial in a specific subgroup if the duration category encompassed the median and the entire interquartile range (IQR) reported in the study paper.

Information sources & search

Pre-existing trials

Randomized clinical trials included in the previous systematic reviews (AlFaleh, 2014; Yang et al., 2014) (hereafter referred to as the “old trials”) were included in this review. Chinese language studies were translated to complete the data extraction (Ke, Su & Li, 2008; Huang, Yang & Huang, 2009; Ren, 2010; Hua, Tang & Mu, 2014; Yang, Yi & Gan, 2011). The studies by Romeo et al. (2011) and Underwood et al. (2009) were divided into two separate trials due to multiple arms.

Updated search

Trials published after completion of the two previous systematic reviews (hereafter referred to as the “new trials”) were identified by searches of Medline, Embase, and the Cochrane Central Register of Controlled Trials. The search was developed and conducted by one of the authors. See Appendix A for the detailed search strategies for the three databases used in this review. Limits were applied to obtain trials from 2013 onwards. No language restrictions were applied.

Updated searches were conducted January 19, 2015. Clinical trial registries were searched on January 14, 2015. Abstracts and conference proceedings were searched on January 15, 2015. On June 3, 2016 another full update of our search strategy was conducted.

We searched for ongoing, unpublished, and terminated trials using the National Library of Medicine and National Institutes of Health clinical trials database and the World Health Organization International Clinical Trials Registry Platform (National Library of Medicine, 2015; World Health Organization International Clinical Trials Registry Platform, 2015). Other sources included electronically available conference materials (2016) from the Society of Pediatric Research (SPR) and the European Society of Pediatric Research (ESPR) (American Pediatric Society/Society for Pediatric Research, 2015; European Society of Pediatric Research, 2015).

Study selection

After de-duplication, two reviewers independently screened titles and abstracts for inclusion using a standardized screening tool. Full text screening was completed independently in duplicate by two authors using a full-text screening tool. Cohen’s Kappa was used to assess agreement between the two reviewers on the selection of full-text articles for inclusion (Freelon, 2015).

Data collection process & data Items

A standardized data collection form was developed a priori and two authors independently extracted the relevant outcomes and validity criteria from the new trials. The data pertaining to the old trials, including risk of bias assessment, was extracted by one author. Disagreements were resolved by consensus and a third party was consulted if necessary. Author contact was attempted for outcome data in the included trials which was missing or unclear. The complete list of the data extracted from the included trials is included in Appendix B.

Study outcome data published in duplicate was included once, but all versions of the publication were utilized for maximal data extraction. In the event of inconsistency between multiple reports of one study, the peer-reviewed publication was used as the primary data set.

Risk of bias within and across studies

Two authors independently assessed the risk of bias for each of the new included studies using the criteria outlined in the Cochrane Handbook for Systematic Reviews of Interventions (Higgins, Deeks & Altman, 2011). A summary table and a graph for risk of bias were created using Review Manager (RevMan) software (The Cochrane Collaboration, 2014). The risk of bias asessments from the studies included in the previous systematic reviews (AlFaleh, 2014; Yang et al., 2014) were double-checked for accuracy by a single author.

Synthesis of results

When possible, the results were synthesized using RevMan 5.3 (The Cochrane Collaboration, 2014). A random effects model (Higgins, Deeks & Altman, 2011) was chosen to account for the clinical and statistical heterogeneity expected when including different species and regimens of probiotics, different neonatal ages and weights, different feeding regimens (breast milk, formula, combination feeding, and parenteral nutrition supplementation as needed), as well as the varied countries conducting RCTs in this area. Relative risks (RRs) with 95% confidence intervals (CIs) were used for dichotomous variables and mean differences (MDs) with 95% CIs for continuous variables. If the continuous variables in the studies were measured in different scales, we calculated the standardized mean difference (SMD).

Analysis was done on an intention-to-treat (ITT) basis (Higgins, Deeks & Altman, 2011). If patients discontinued the intervention after randomization, they were still counted in our analysis for outcomes (such as mortality) where this was possible. Author contact was attempted to clarify any missing outcome data.

If trials had two intervention arms, both of which contained a probiotic, both probiotic arms were included and the number of patients in the comparator arm divided by the number of active arms to prevent double counting. If the trial had two or more intervention arms and only one of them contained a probiotic, the data from the corresponding non-probiotic arm was used as the comparator (Higgins, Deeks & Altman, 2011). In trials where patients received a co-intervention, the co-intervention had to be present in both the active and control arms to be included.

The I2 statistic was used to quantify statistical heterogeneity (the percentage of total variation across studies due to heterogeneity). Statistical heterogeneity as measured by I2 was described as “small” (≤25%), “moderate” (between 26% and 49%) and “large” (≥50%) (Higgins et al., 2003). Forest plots were visually inspected for possible sources of heterogeneity.

Additional analysis

Subgroup analysis was planned a priori for the following subgroups: infant weight (extremely-low birth weight (ELBW) (less than 1,000 g) and very low birth weight (VLBW) (less than 1,500 g)), timing of probiotic initiation, duration of probiotic therapy, sepsis types (including “any sepsis”), and use of breastmilk vs. formula for feeding.

Results

Study selection

The previously published systematic reviews included a total of 37 unique randomized clinical trials (Ke, Su & Li, 2008; Huang, Yang & Huang, 2009; Ren, 2010; Yang, Yi & Gan, 2011; Romeo et al., 2011; Underwood et al., 2009; Al-Hosni et al., 2012; Bin-Nun et al., 2005; Braga et al., 2011; Costalos et al., 2003; Dani et al., 2002; Demirel et al., 2013; Fernandez-Carrocera et al., 2013; Kitajima et al., 1997; Li et al., 2004; Lin et al., 2005; Lin et al., 2008; Manzoni et al., 2006; Manzoni et al., 2009; Mihatsch et al., 2010; Millar et al., 1993; Mohan et al., 2006; Jacobs et al., 2013; Reuman et al., 1986; Rojas et al., 2012; Rougé et al., 2009; Samanta et al., 2009; Sari et al., 2011; Stratiki et al., 2007). Electronic database searches (including the 2016 search update) yielded 475 citations, conference searching yielded 115 citations, and clinical trials database searching yielded 35 citations. After de-duplication, 412 citations remained for title and abstract screening (see Fig. 1 for the detailed flow diagram of study selection). Cohen’s Kappa was 0.723 (good agreement) between the two reviewers for selection of new full-text trials for inclusion (Altman, 2001). The study by Manzoni et al. (2014) was included in our review as it was a randomized extension of a previously published trial (Manzoni et al., 2009). The ProPrems study was added to the previous review as unpublished data, but is now published and was included in our review (Jacobs et al., 2013). The updated search in 2016 resulted in a follow-up to the Oncel trial (Oncel et al., 2014) (Akar; Akar et al., 2016), and new trials by Costeloe (Costeloe et al., 2016; previously on our ongoing trials list), Dilli (Dilli et al., 2015), Dutta (Dutta, Ray & Narang, 2015), Sinha (Sinha et al., 2015) and Tewari (Tewari, Dubey & Gupta, 2015). Three trials included in previous reviews were excluded from our review as they were determined to be non-randomized (Fu & Song, 2012; Hunter et al., 2012; Li, Qiao & Huang, 2011). The overall updated search added a total of 13 randomized controlled trials (two trials split due to multiple arms) with over 5,000 new evaluable patients to previous systematic reviews, bringing the total to 42 included trials (Hua, Tang & Mu, 2014; Manzoni et al., 2014; Oncel et al., 2014; Costeloe et al., 2016; Dilli et al., 2015; Dutta, Ray & Narang, 2015; Sinha et al., 2015; Tewari, Dubey & Gupta, 2015; Patole et al., 2014; Roy et al., 2014; Serce et al., 2013; Van Niekerk et al., 2014a).

Figure 1 PRISMA Flow Diagram.

Study Characteristics

When verifying the outcome data included in the previous reviews, a number of methodological flaws and errors of data synthesis were noted (AlFaleh, 2014; Yang et al., 2014). A decision was made to re-extract the data from the “old trials” instead of re-entering the data from the previously published reviews (see Appendix C).

See Table 1 for characteristics of included studies. All studies were conducted in preterm infants admitted to the NICU. Twenty-four studies limited birth weight to 1,500 g or less. Weight was not part of the inclusion criteria in nine studies (Ke, Su & Li, 2008; Hua, Tang & Mu, 2014; Costalos et al., 2003; Millar et al., 1993; Mohan et al., 2006; Stratiki et al., 2007; Costeloe et al., 2016; Dutta, Ray & Narang, 2015; Tewari, Dubey & Gupta, 2015). Gestational age was not part of the inclusion criteria in five studies but all of these studies had birth weight inclusion criteria for preterm infants less than 1,500 g (Braga et al., 2011; Kitajima et al., 1997; Lin et al., 2005; Manzoni et al., 2006; Manzoni et al., 2014). One trial did not specify gestational age but enrolled babies 1,500–2,500 g (Sinha et al., 2015). Five trials were translated from Chinese for use in the review (Ke, Su & Li, 2008; Huang, Yang & Huang, 2009; Ren, 2010; Hua, Tang & Mu, 2014; Li et al., 2004).

Table 1 Characteristics of Included Studies.

	Inclusion Criteria							
Identifier	Gestational age	Birth weight	Other inclusion criteria	Number randomized in each group	Probiotic Species (Brand names)	Total Dose (cfu/day)	Initiationa	Durationb	Feeding (B, PF, F, Mixed)	
Al-Hosni et al. (2012)	“preterm”	501–1,000 g	14 days of age or less at the time of initiation of feeds	Probiotic: 50	Lactobacillus rhamnosus GG LGG—0.5 billion (Culturelle®)	1 billion	At the time of first feeding	28 days or more	Not stated	
Control: 51	Bifidobacterium infantis—0.5 billion (Align®)	
Bin-Nun et al. (2005)	“preterm”	Less than 1,500 g	None	Probiotic: 72	B. infantis—0.35 billion	1.05 billion	At the time of first feeding	28 days or more	Mixed	
Bifidobacteria bifidus—0.35 billion	
Control: 73	Streptococcus thermophilus—0.35 billion (ABC Dophilus®)	
Braga et al. (2011)	None	750–1,499 g	Born locally and admitted to NICU	Probiotic: 119	Lactobacillus casei—0.002 to 2 billion	0.035–3.5 billion	48 h or less	Planned for 30 d of life, diagnosis of NEC, discharge from hospital or death, whichever occurred first	Mixed	
Control: 112	Bifidobacterium breve—0.005 to 5 billion(Yakult LB®- Sao Paulo, Brazil)	
Costalos et al. (2003)	28–32 weeks	None	None	Probiotic: 51	Saccharomyces boulardii	2 billion	At the time of first feeding	28 days or more	Mixed	
Control: 36	
Costeloe et al. (2016)	23 weeks up to 30 weeks and 6 days	None	None	Probiotic:650	B. breve BBG-001 (Yakult Honsha Co Ltd –Tokyo, Japan)	0.067–6.7 billion	48 h or less	28 days or more	Mixed /B (46%)	
Control: 660	
Dani et al. (2002)	Less than 33 weeks	Less than 1,500 g	None	Probiotic: 295	L. rhamnosus GG (Dicoflor®)	6 billion	At the time of first feeding	28 days or more	Mixed	
Control: 290	
Demirel et al. (2013)	Less than 32 weeks	1,500 g or less	Survival to start enteral feeding	Probiotic: 135	S. boulardii (Reflor®)	5 billion	At the time of first feeding	28 days or more	Mixed	
Control: 136	
Dilli et al. (2015)c	Less than 32 weeks	Less than 1,500 g	7 days of age or less at the time of initiation of feeds	Probiotic: 100		5 billion	More than 48 h	28 days or more	Mixed	
Synbiotic: 100	B. lactis 5 billion	
Prebiotic: 100	B. lactis 5 billion + inulin	
Control: 100	Inulin 900 mg (Maflor®)	
Dutta, Ray & Narang (2015)	27–33 weeks	None	Aged less than 96 hrs, likely to remain in hospital or reside within 30 km for 28 days, tolerating 15 mL/kg/d of milk feeds	High-dose long course: 38	Low Dose: L. acidophilus (662.5 million), L. rhamnosus (362.5 million), B. longum (87.5 million), S. boulardii (137.5 million); High Dose: L. acidophilus (5.3 billion), L. rhamnosus (2.9 billion), B. longum (700 million), S. boulardii (1.1 billion)		Within the first week	28 days or more	Mixed	
High-dose short course: 38	Low dose: 2 billion	
Low dose, long course: 38	High dose: 20 billion	
Control: 35		
Fernandez-Carrocera et al. (2013)	“preterm”	Less than 1,500 g	None		Lactobacillus acidophilus—1 billion	2.65 billion	At the time of first feeding	28 days or more	Mixed	
	L. rhamnosus—0.44 billion	
	L. casei—1 billion	
Probiotic: 75	Lactobacillus plantarum—0.176 billion	
Control: 75	B. infantis—0.0276 billion	
	S. thermophillus—0.0066 billion	
	(Lactipan®)	
Hua, Tang & Mu (2014)	Less than 37 weeks	None	Anticipated to start enteral feeding within 72 hrs.	Probiotic: 119	Bifidobacterium longum	3 billion	At the time of first feeding	14–27 days	Mixed	
		Lactobacillus bulgaricus	
Anticipated length of stay at least 7 days.	Control: 138	S. thermophiles (Golden Bifid®)	
Huang, Yang & Huang (2009)	28–32 weeks	Less than 1,500 g	None	Probiotic: 95	Bifidobacterium adolescentis	0.05 billion	More than 48 hrs	Up to 13 days	Unknown	
Control: 88	
Jacobs et al. (2013) (ProPrems)	Less than 32 weeks	Less than 1,500 g	Enrolled within 72 h of birth.	Probiotic: 548	B. infantis—0.35 billion	1 billion	More than 48 hrs	28 days or more	Mixed	
B. bifidus—0.35 billion	
Control: 551	S. thermophilus—0.35 billion (ABC Dophilus®)	
Ke, Su & Li (2008)	Less than 37 weeks	None	None	Probiotic: 438	Enterococcus faecalis—1 billion	3 billion	More than 48 hrs	Until Discharge	Unknown	
	B. longum—1 billion	
Control: 446	L. acidophilus—1 billion (Bifico®)	
Kitajima et al. (1997)	None	Less than 1,500 g	None	Probiotic: 45	B. breve YIT4010 (Yakult®Honsya Co. Ltd., Tokyo, Japan)	0.5 billion	At the time of first feeding	28 days or more	Mixed	
Control: 46	
Li et al. (2004)	27.8–37.6 weeks	780–2,250 g	Stated as low birth weight infants	Probiotic: 20	B. breve	0.32 billion	48 h or less	Until Discharge	Unknown	
Control: 10	
Lin et al. (2005)	None	Less than 1,500 g	None	Probiotic: 180	L. acidophilus—1 billion/250 mg cap	1 billion/kg	At the time of first feeding	28 days or more	B	
Control: 187	B. infantis—1 billion/250 mg cap (Infloran®- Laboratorio Farmaceutico, Italy)	
Lin et al. (2008)	Less than 34 weeks	Less than 1,500 g	None	Probiotic: 222	L. acidophilus—1 billion/250 mg cap	1 billion/kg	At the time of first feeding	Until Discharge	Mixed	
Control: 221	B. bifidum—1 billion/250 mg cap (Infloran®- Laboratorio Farmaceutico, Italy)	
Manzoni et al. (2006)	None	Less than 1,500 g	Less than 3 days of age, started oral feeding with human milk, no baseline fungal colonization at enrollment, no other antifungal prophylaxis, oral feeding was stable and was tolerated by neonate	Probiotic: 39	L. rhamnosus GG (Dicoflor®)	6 billion	More than 48 hrs	28 days or more	B	
Control: 41	
Manzoni et al. (2014)d	None	Less than 1,500 g	Less than 48 h of age	Synbiotic: 238	L. rhamnosus GG 6 billion + Bovine Lactoferrin 100 mg (Dicoflor®)	6 billion	More than 48 hrs	28 days or more	Mixed	
Prebiotic: 258	Bovine Lactoferrin 100 mg (Dicofarm®)	
Mihatsch et al. (2010)	Less than 30 weeks	Less than 1,500 g	None	Probiotic: 93	Bifidobacterium lactis—20 billion/g (Nestle®)	12 billion/kg	At the time of first feeding	28 days or more	Mixed	
Control: 90	
Millar et al. (1993)	33 weeks or less	None	None	Probiotic: 10	L. rhamnosus GG (Valio Finnish Co-operative Dairies Association®)	0.2 billion	At the time of first feeding	14 days	Mixed	
Control: 10	
Mohan et al. (2006)	Less than 37 weeks	None	None	Probiotic:37	B. lactis Bb12—2 billion/g (Nestle FM 2000A®)	4.8 billion	48 h or less	14–27 days	F/B status not stated	
Control: 32	
Oncel et al. (2014)	32 weeks or less	1,500 g or less	None	Probiotic: 200	Lactobacillus reuteri DSM 17938 in oil (Biogaia®)	0.1 billion	At the time of first feeding	28 days or more	Mixed	
Control: 200	
Patole et al. (2014)	Less than 33 weeks	Less than 1500 g	Ready to commence or on enteral feeds for <12 h	Probiotic: 77	B. breve M-16V (Morinaga Milk Industry Co, Ltd®, Tokyo, Japan)	3 billion	At the time of first feeding	28 days or more	Mixed	
Control: 76	
Ren (2010)	28–33 weeks	1,000–1,800 g	None	Probiotic: 80	B. infantis—0.005 billion	0.016 billion	At the time of first feeding	Up to 13 days	Unknown	
	L. acidophilus—0.005 billion	
Control: 70	E. faecalis—0.005 billion	
	Bacillus cereus—0.0005 billion (Bifidobacterium tetravaccine)	
Reuman et al. (1986)	“preterm”	Less than 2,000 g	Greater than 24 hrs, but less than 72 hrs old	Probiotic: 15	L. acidophilus (Chris Hansen Laboratory, Inc.®, Milwaukee, WI)	0.018 billion	Within 72 hrs	28 days or more	Mixed	
Control: 15	
Rojas et al. (2012)	“preterm”	2,000 g or less	None	Probiotic: 372	L. reuteri DSM 17938 in oil (Biogaia®)	0.1 billion	48 h or less	14–27 days	Mixed	
Control: 378	
Romeo et al. (2011)e	Less than 37 weeks	Less than 2,500 g	–age < 2wks	Probiotic (L. reuteri): 83	L. reuteri DSM 17938 in oil (Biogaia®)	0.1 billion L. reuteri or	More than 48 hrs	14–27 days	Mixed	
–feeds within 72 hrs	Probiotic (L. rhamnosus): 83	L. rhamnosus GG (Dicoflor®)	6 billion L. rhamnosus	
	Control: 83			
Rougé et al. (2009)	Less than 32 week	Less than 1,500 g	postnatal age <∕ = 2 week, the absence of any disease other than those linked to prematurity and the start of enteral feeding	Probiotic:43	L. rhamnosus GG—0.1 billion (Valio, Ltd®)	0.8 billion	At the time of first feeding	28 days or more	B	
Placebo: 49	B. longum BB536—0.1 billion (Morinaga Milk Industry Co, Ltd®, Tokyo, Japan)	
Roy et al. (2014)	Less than 37 weeks	Less than 2,500 g	Stable oral feeding within 72 h of birth, adequate renal and liver function, a postnatal age <2 week	Probiotic: 56	L. acidophilus—1.25 billion/g	1.25 billion	More than 48 hrs	28 days or more	B	
	B. longum—0.125 billion/g	
	B. bifidum—0.125 billion/g	
Control: 56	B. lactis—1 billion/g	
	(Prowel®)	
Samanta et al. (2009)	Less than 32 weeks	Less than 1,500 g	Started feed enterally and survived beyond 48 h of life	Probiotic: 91	B. infantis—2.5 billion	20 billion	More than 48 hrs	14–27 days	B	
	B. bifidum—2.5 billion	
Control: 95	B. longum—2.5 billion	
	L. acidophilus—2.5 billion	
Sari et al. (2011)	Less than 33 weeks	Less than 1,500 g	who survived to feed enterally	Probiotic: 110	Bacillus coagulans (Lactobacillus sporogenes)	0.35 billion	At the time of first feeding	28 days or more	Mixed	
Control: 111	(DMG ITALIA SRL®, Rome, Italy)	
Serce et al. (2013)	32 weeks or less	1,500 g or less	Survival to feed enterally	Probiotic: 104	S. boulardii (Reflor®)	1 billion	At the time of first feeding	28 days or more	Mixed	
Control: 104	
Sinha et al. (2015)	None	1,500–2,500 g	Residing within 20–25 km of hospital and not planning to shift residences for at least the next 2 months	Probiotic: 668	VSL#3®: Streptococcus thermophilus, Bifidobacterium breve, Bifidobacterium longum, Bifidobacterium infantis, Lactobacillus acidophilus, Lactobacillus plantarum, Lactobacillus paracasei and Lactobacillus delbrueckii spp bulgaricus.	10 billion	Within the first week	28 days or more	B	
Control: 672	
Stratiki et al. (2007)	27–37 weeks	None	formula fed	Probiotic: 41	B. lactis (Prenan Nestlé®)	0.2 billion/kg	48 h or less	Not stated	PF	
Control: 34	
Tewari, Dubey & Gupta (2015)	27–30 weeks + 6 days and 31–33 weeks + 6 days	None	None	Probiotic:123	Bacillus clausii 2 billion (Enterogermina®)	6 billion	More than 48 hrs	28 days or more	B	
Control:121	
Underwood et al. (2009)e	Less than 35 weeks	750–2,000 g	Younger than 7 days old	Probiotic: (Culturelle): 30	L. rhamnosus GG—10 billion/cap	0.5 billion Culturelle or	Within the First Week	28 days or more	Mixed	
	(ProBioPlus DDS)		
	B. infantis—10 billion/cap		
Probiotic: (ProBioPlus): 31	B. bifidum—10 billion/cap	2 billion ProBioPlus	
	B. longum—10 billion/cap		
Control: 29	L. acidophilus—10 billion/cap		
	(Culturelle®)		
Van Niekerk et al. (2014a); Van Niekerk et al. (2014b)f	Less than 34 weeks	500–1,250 g	HIV exposed and unexposed born to HIV positive or negative mothers who agreed to breastfeed	Probiotic: 91	L. rhamnosus GG—0.35 billion	0.7 billion	At the time of first feeding	28 days or more	B	
Control: 93	B. infantis—0.35 billion	
	(Pro-B2®)	
Yang, Yi & Gan (2011)	Less than 37 weeks	<1,500–>2,500 g	2 week length of stay and admitted within 24 h	Probiotic: 31	B. longum—0.005 billion	0.03 billion	At the time of first feeding	Up to 13 days	Unknown	
	L. acidophilus—0.005 billion	
Control: 31	E. faecalis—0.005 billion	
Notes.

a Initiation of probiotic therapy was categorized to fit the defined subgroups for data analysis.

b Duration of probiotic therapy was categorized to fit the defined subgroups for data analysis.

c Handled as two trials (4 arms).

d Randomized extension of the 2009 publication (Lin et al., 2005).

e Handled as two trials to account for the 3 arms in the trial.

f Included two randomized clinical studies, one of HIV-exposed and one of HIV-unexposed preterm infants which were analyzed as two trials.

B Breastfeeding only

PF Preterm formula

F Formula

Mixed Mixed feeding types

Type of feeding was variable across the included trials. Nine trials included infants exclusively fed breastmilk (Lin et al., 2005; Manzoni et al., 2006; Rougé et al., 2009; Samanta et al., 2009; Sinha et al., 2015; Tewari, Dubey & Gupta, 2015; Roy et al., 2014; Van Niekerk et al., 2014b). One trial had infants fed exclusively preterm formula (Stratiki et al., 2007). The trials published in Chinese did not consistently specify this information on translation (Ke, Su & Li, 2008; Huang, Yang & Huang, 2009; Ren, 2010; Yang, Yi & Gan, 2011; Li et al., 2004). Costeloe et al. (2016) had 46% of infants exclusively fed breastmilk, but the rest of the infants had a combination of feeding types.

Overall, the number of trials were split evenly between multiple species and single species probiotics (22 trials each). The Sari trial (Bacillus coagulans formerly known as Lactobacillus sporogenes) (Sari et al., 2011) and the Tewari trial (Bacillus clausii) (Tewari, Dubey & Gupta, 2015) used single species that were not used in any other trial. Sinha used a multi-organism product containing eight species (Sinha et al., 2015). All studies used a variety of organisms and dose regimens. Comparators were matching placebo, standard therapy, or prebiotics (two trials) (Manzoni et al., 2014; Dilli et al., 2015). There were no trials comparing one probiotic preparation with another, but two trials had multiple arms with different probiotics (Romeo et al., 2011; Underwood et al., 2009). One trial used varying durations of probiotics and doses but fit within the range of doses and duration of therapy seen with all included trials, so the three treatment arms were combined into one (Dutta, Ray & Narang, 2015).

Timing of probiotic initiation was variable. Twenty-one trials started probiotics with the first feed, six trials started within 48 h of birth, one within 72 h, four within the first week, and in twelve trials therapy started at the “more than 48 h” time point.

Duration of probiotic therapy ranged from seven days to six weeks. One trial did not specify a duration of therapy (Stratiki et al., 2007). Most studies were classified in the “28 days or more” subgroup for the purposes of analysis by extraction of the actual duration of therapy (when provided) in trials that specified duration as “until discharge.”

Outcomes

Risk of Bias within Studies.

See Fig. S1 for the risk of bias assessment for all included trials. All included trials were randomized (five were judged to have uncertainty around the method of randomization) (Al-Hosni et al., 2012; Bin-Nun et al., 2005; Dani et al., 2002; Kitajima et al., 1997; Reuman et al., 1986). All of these trials were previously included in the AlFaleh review. Seven trials had a degree of selective reporting one of the trials being from the updated search (Roy et al., 2014). Of the translated trials, randomization was clearly stated, but uncertainty remains about blinding status, allocation concealment, and selective reporting (Ke, Su & Li, 2008; Huang, Yang & Huang, 2009; Ren, 2010; Hua, Tang & Mu, 2014; Li et al., 2004).

Synthesis of results.

Two of the “old trials” did not contribute any outcome data to the meta-analysis and were excluded (Li et al., 2004; Mohan et al., 2006). Data used for the Mohan trial in the previous review appears to be based on personal communication with the authors and could not be corroborated with the published trial (Mohan et al., 2006). Li did not report on any usable outcomes (Li et al., 2004).

All infants

The primary outcome, severe NEC, was significantly reduced in infants who received probiotics compared to placebo with 38 trials (10,520 patients) reporting on this outcome—RR 0.53 95% CI [0.42–0.66]—see Fig. 2. The incidence of culture-proven sepsis was not different between the probiotics and control—RR 0.88 95% CI [0.77–1.00] in 31 trials comprising 8,707 patients, see Fig. 3. The incidence of all-cause mortality was significantly reduced in infants receiving probiotics in 29 trials (9,507 patients)—RR 0.79 95% CI [0.68–0.93] (Fig. 4). Other statistically significant findings included shorter duration of hospitalization, increased weight gain (g/day), and reduced time to reach full enteral feeds, all in favor of using probiotics (Table 2).

There was a moderate to large degree of heterogeneity in the results for culture-proven sepsis, duration of hospitalization, duration of parenteral nutrition, and time to achieve full feeds.

Figure 2 Forest plot showing the effect of probiotics on severe NEC in all infants.

Figure 3 Forest plot showing the effect of probiotics on culture-proven sepsis in all infants.

Figure 4 Forest plot showing the effect of probiotics on all-cause mortality in all infants.

Table 2 Additional important findings.

Outcome	Number of studies / participants	Effect size	95% CI	I2 (%)	
All Infants	
Bacterial sepsis	9 / 2212	RR 0.86	0.62 to 1.18	52	
Fungal sepsis	12 / 3756	RR 0.67	0.43 to 1.06	10	
Duration of hospitalization (days)	16 / 4915	MD −3.2	−5.5 to −0.9	84	
Weight gain (g/day)	3 / 314	MD +1.7	1.0 to 2.3	0	
Time to achieve full feeds (days)	17 / 4448	MD −1.2	−2.2 to −0.1	93	
VLBW infants	
Culture-proven sepsis	24 / 6616	RR 0.93	0.82 to 1.05	15	
Duration parenteral nutrition (days)	4 / 1210	MD −1.2	−2.3 to −0.02		
ELBW infants	
Culture-proven sepsis	6 / 1703	RR 0.95	0.72 to 1.26	41	
Mortality	4 / 1122	RR 0.92	0.046 to 1.83	47	
Duration of hospitalization (days)	2 / 218	MD −6.4	−12.6 to −0.1		
Time to achieve full feeds (days)	2 / 218	MD −1.8	−2.9 to −0.7		
Notes.

MD Mean difference

RR Risk ratio

CI Confidence interval

NEC Necrotizing enterocolitis

VLBW Very low birth weight (<1,500 g)

ELBW Extremely low birth weight (<1,000 g)

VLBW infants

The incidence of severe NEC was significantly reduced in VLBW infants who received probiotics compared to placebo including 25 trials (6,587 patients)—RR 0.47 95% CI [0.36–0.61] (Fig. 5). The incidence of all-cause mortality was significantly reduced in VLBW infants who received probiotics compared to infants who received placebo in 24 trials (6736 patients) with RR 0.74 95% CI [0.61–0.90]. Compared to VLBW infants who received placebo, those who received probiotics had a significantly reduced duration of parenteral nutrition (Table 2).

There was significant heterogeneity in the outcomes of duration of hospitalization, and time to full feeds.

Figure 5 Forest plot showing the effect of probiotics on NEC in VLBW infants.

ELBW infants

Eight trials reported outcome data on this weight group (Al-Hosni et al., 2012; Lin et al., 2008; Manzoni et al., 2006; Manzoni et al., 2009; Jacobs et al., 2013; Oncel et al., 2014; Costeloe et al., 2016; Roy et al., 2014). The only trial to enroll infants solely in this weight group was Al-Hosni et al. (2012). ELBW infants were a pre-specified subgroup in the Jacobs trial (Jacobs et al., 2013). In the remaining six trials, outcome data for ELBW infants was presented as a post-hoc subgroup analysis. ELBW infants who received probiotics had a significantly shorter duration of hospitalization and reached full enteral feeding sooner compared to infants who received placebo, see Table 2.

No statistically significant differences were demonstrated for the incidence of NEC (Fig. 6), mortality, culture-proven sepsis, any bacterial sepsis and any fungal sepsis. There was significant heterogeneity in the outcomes of culture-proven sepsis and mortality. Other outcomes were only reported in a small number of patients and trials.

Figure 6 Forest plot showing the effect of probiotics on NEC in ELBW infants.

Initiation of probiotics

Severe NEC was significantly reduced in trials where patients were started on probiotics at more than 48 h of age—RR 0.36 95% CI [0.24–0.53] or in those trials where probiotics were started at the time of the first feed—RR 0.55 95% CI [0.41–0.75] (Fig. S2). The incidence of culture-proven sepsis was significantly reduced in the 11 trials in which therapy was started at more than 48 h of age—RR 0.65 95% CI [0.51–0.82]. A reduction in the incidence of mortality was significant in trials when probiotics were started with the first feed—RR 0.68 95% CI [0.51–0.90].

Duration of probiotics

Subgroups with probiotic duration of at least 14 days or until discharge were statistically significant for a reduced incidence of severe NEC (Fig. S3). The largest amount of data was in the 28 days or more category, with 28 trials contributing outcome data.

Species of probiotics

Outcomes were compared according to the various probiotic species included in the trials. Incidence of severe NEC was significantly reduced in infants receiving a Lactobacillus species (8 trials)—RR 0.61 95% CI [0.40–0.95], Bifidobacterium species (6 trials)—RR 0.37 95% CI [0.14–0.97], or multispecies (two or more) supplement (18 trials)—RR 0.41 95% CI [0.29–0.56]. Incidence of NEC was not significantly different from control in infants receiving only a Saccharomyces boulardii supplement (2 trials)—RR 0.72 95% CI [0.33–1.54]. Incidence of culture-proven sepsis was not significantly different from control in infants receiving any probiotic species. Incidence of mortality was significantly reduced only in infants receiving a multispecies supplement (15 trials)—RR 0.66 95% CI [0.5–0.87].

Breastmilk vs. formula feeding

Comparison of the rates of severe NEC between infants fed using breast milk alone and those fed formula alone was not possible due to the lack of studies containing infants fed only formula.

Discussion

This review was done in accordance with current guidelines and strict attention to best practice of systematic reviews and meta-analysis (Moher et al., 2009). It has added randomized data from over 5,000 infants to the previous meta-analyses. Based on high-quality evidence, the use of probiotics in preterm infants reduces the incidence of severe NEC. The effect size has changed slightly in comparison to the Cochrane review but the precision of the result remains the same, despite the additional patients (AlFaleh, 2014). This may be related to the wide range of probiotic species and regimens included in the analysis and use of the more conservative random effects model for meta-analysis. There was no statistical heterogeneity in the primary outcome, despite the inclusion of diverse probiotic regimens and species. No other intervention to prevent NEC has demonstrated this effect size (Foster, Seth & Cole, 2004; Pammi & Abrams, 2015; Bury & Tudehope, 2001).

This review showed a decrease in all-cause mortality with probiotics, which confirms the findings of previous reviews and re-affirms the important benefit of this therapy.

The concern about bacterial translocation beyond the preterm infant gut should be reflected in the outcome of culture-proven sepsis and/or all-cause mortality. This review found no increased risk of culture-proven sepsis. No sepsis due to probiotic species was reported among the included trials.

A statistically significant reduction of three days was shown in duration of hospitalization. The clinical significance of this reduction is unclear given a mean length of stay in Canadian NICUs of 63.2 days in 2013 (The Canadian Neonatal Network, 2013).

The reduction in the duration of parenteral nutrition and time to full enteral feeds is of importance for this patient population, as prolonged parenteral nutrition may be associated with increased hospital stay, mortality, and morbidity (Flidel-Rimon et al., 2004). Recently published evidence-based guidelines echo the need and benefits of achieving full feeds in an efficient manner (Dutta et al., 2015).

In the ELBW infants, the lack of benefit on severe NEC, culture-proven sepsis or mortality outcomes was consistent with the previous reviews (despite the addition of four new randomized trials almost doubling the number of infants studied). The direction and magnitude of the point estimates for the effect of probiotics on the incidence of severe NEC and all-cause mortality are consistent with those of the “all infant” sample.

The incidence of NEC and mortality outcomes had little to no heterogeneity which gives substantial confidence in those results. The substantial heterogeneity in sepsis, duration of hospitalization and duration of parenteral nutrition outcomes would suggest caution in interpreting the results.

Timing of probiotic initiation is a clinically important question which was not resolved in the previous reviews. In this review, subgroups for timing mirrored those in the Alfaleh review (AlFaleh, 2014). The time of initiation of probiotics seemed to have a variable influence on the main three outcomes of severe NEC, culture-proven sepsis, and mortality. When probiotics were started very early (48 h of age or less) there was no difference in any of the outcomes. There were few trials placed in this category, and therefore the outcomes may lack power to detect a statistical difference. Many trials described initiating probiotic supplementation at the time of first feeding. Without access to individual patient level data, it is unclear how many of the infants categorized into this group could also be included in the 48 h of age or less category. Consequently, we cannot definitively state that probiotic supplementation should be withheld until at least 48 h of age or until feeding. Starting probiotics with the initiation of feeds did reduce the incidence of both NEC and mortality and does have some practical advantages in terms of drug administration which make it an opportune time to initiate probiotic prophylaxis. There was a lack of effect on mortality when probiotic supplementation was started after 48 h of age. We can find no explanation for this, especially since the benefit on NEC remained when therapy was started after 48 h.

Determining the appropriate duration of therapy is equally important as the timing of initiation. Clinically it seems prudent to continue therapy for as long as there is risk for NEC. A minimum of two weeks of probiotic therapy continued for as long as the patient is judged to be at risk (up to six weeks) can be recommended, since trials in these subgroups showed a lower incidence of NEC.

Feeding infants with human milk compared to formula has been previously shown to have a protective effect on the incidence of NEC (Sullivan et al., 2010; Meinzen-Derr et al., 2009). This review found only one trial in which infants were fed exclusively formula (most other trials included a combination of feeding types), precluding definitive conclusions based on feeding method. The majority of infants were fed a combination of human milk and formula reflecting clinical practice. Future trials may consider having a pre-defined subgroup of breastfed vs. formula fed infants to definitively answer this question.

A post hoc subgroup analysis to examine if the effects on severe NEC were consistent based on the underlying background incidence of NEC across the included trials (grouped by less than 5%, 5–7% and more than 7% (Henry & Moss, 2008)) was undertaken. Most of the trials were in the low baseline incidence subgroup (18 trials, 4,905 patients). The primary outcome remained significant across all groups and reinforces that no matter the institution’s incidence of NEC, infants had the same reduction in severe NEC.

In many countries, probiotics are not regulated as drugs and products are not subject to the same rigorous quality assurance standards (Venugopalan, Shriner & Wong-Beringer, 2010). Stability and/or species testing was confirmed in nine of the included trials (Underwood et al., 2009; Al-Hosni et al., 2012; Fernandez-Carrocera et al., 2013; Mihatsch et al., 2010; Millar et al., 1993; Patole et al., 2014; Van Niekerk et al., 2014a; Van Niekerk et al., 2015). Hospitals either did their own testing or requested the information from the manufacturer of the probiotic being studied. Institutions are encouraged to conduct their own quality assessment or request quality certificates from the manufacturer of the product being used (Chan, Soltani & Hazlet, 2015; Barrington & Janvier, 2015).

Limitations

The limitations to this systematic review were as follows:

1. Three of the Chinese language trials (Di & Li, 2010; Deng & Chen, 2010; Zhou, 2012) included in the older review (Yang et al., 2014) could not be obtained in full text and were not included in this review.

2. No unpublished data was requested from any of the manufacturers of probiotic products assessed in this review.

3. Only one trial in the previous review addressed long term neurodevelopmental outcomes, but this information could not be confirmed (Kitajima et al., 1997). Akar 2016 (Akar et al., 2016) and the abstract from one of the ProPrems conference presentations (Jacobs et al., 2016) also reports on neurodevelopmental outcomes. If the Kitajima (Kitajima et al., 1997) and ProPrems results were available these could be combined for a summary effect estimate in a future review.

Remaining uncertainties

The outcome of fungal sepsis showed a definite benefit with no heterogeneity (Table 2). Some of the included studies employed antifungal prophylaxis (either systemic or topical) in their infants as per their normal NICU practice. This choice is not the routine practice at all institutions and is not standard practice (Benjamin jr et al., 2014; Austin, Darlow & McGuire, 2013). The impact of these studies with background antifungal therapy was not explored in sensitivity analyses but could be considered in future reviews for its impact on the outcome of fungal sepsis.

Which probiotic product to use remains uncertain, since the total body of evidence comprises a heterogeneous group of probiotics (individual species and combination products, and regimens). In the previous review, only the Lactobacillus and multispecies supplements were shown to be effective for this outcome. We would recommend a regulatory body-approved product and that quality assessment be requested from the manufacturer to validate the purity of product. The evidence of benefit was clear for Lactobacillus or Bifidobacterium species and multiple species products so any of these would be reasonable choices.

Conclusions

For infants born at less than 37 weeks gestation or less than 2,500 g birth weight there is clear benefit from the use of probiotics to prevent severe NEC and all-cause mortality, with no increase in culture-proven sepsis. We would recommend using probiotics in premature infants with these characteristics. The evidence for babies of birth weight less than 1,000 grams is less clear and we cannot make as strong a recommendation in this class of infants.

Supplemental Information

Dataset S1 Review Manager (RevMan) file Probiotics in NEC

Dataset used for analysis for meta analysis

Click here for additional data file.

Supplemental Information 1 PRISMA Checklist for Systematic Reviews and Meta Analysis

Click here for additional data file.

Supplemental Information 2 Supplementary Materials

Click here for additional data file.

We gratefully acknowledge the following LHSC pharmacists for their expertise in translating the Chinese language papers: Emily Chen, Rachel Fu, Vicky Luo, and Boris Tong. We thank Brenda Sampson for her efforts in obtaining articles and the LHSC library staff Valerie Kowalkowski and Juanita Meyer in locating articles.

Additional Information and Declarations

Competing Interests

Author Contributions

Data Availability

The authors declare there are no competing interests.

Sonja C. Sawh conceived and designed the experiments, performed the experiments, analyzed the data, contributed reagents/materials/analysis tools, wrote the paper, prepared figures and/or tables, reviewed drafts of the paper.

Santosh Deshpande performed the experiments, analyzed the data, contributed reagents/materials/analysis tools, wrote the paper, prepared figures and/or tables, reviewed drafts of the paper, design and conduct of the search strategy of the electronic databases.

Sandy Jansen conceived and designed the experiments, wrote the paper, prepared figures and/or tables, reviewed drafts of the paper.

Christopher J. Reynaert performed the experiments, contributed reagents/materials/analysis tools, wrote the paper, prepared figures and/or tables, reviewed drafts of the paper, search of the clinical trials databases.

Philip M. Jones conceived and designed the experiments, analyzed the data, contributed reagents/materials/analysis tools, wrote the paper, prepared figures and/or tables, reviewed drafts of the paper, data interpretation.

The following information was supplied regarding data availability:

The raw data has been supplied as a Supplemental File.

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
