# Peer review of "Prevention of necrotizing enterocolitis with probiotics: a systematic review and meta-analysis"

_PeerJ, doi:10.7717/peerj.2429_

## Round 0.1 · original submission · Major Revisions

· Academic Editor

Major Revisions

Please ensure you address all of the reviewers' comments in the revised article, and include a point by point description of the changes that have been made.

Please update the statement in the Introduction on the previous SRs to include the third SR (Aceti 2015) mentioned by Reviewer 2 and to describe how the present paper differs from the existing SRs. Consider expanding on your analysis as suggested by Reviewer 2.

The authors could also consider updating their 14 months old search.

·

Basic reporting

The article is reported in accordance with the journal´s policies and written in a very clear and unambiguous language that conforms to professional standards of courtesy and expression. The background gives a good introduction to the field of necrotizing enterocolitis and probiotics. Literature is appropriately referenced. All figures are relevant to the content of the article, of sufficient resolution, and appropriately described and labeled. The submission includes all results that are relevant. All appropriate raw data has been made available in accordance with the journal´s data sharing policy.

Experimental design

The article describes original primary research in a meta-analysis within the scope of the journal. The research question – primary outcome “severe necrotizing enterocolitis” – is clearly defined and meaningful. The knowledge gap being investigated is an update of two prior recent meta-analyses, which is clearly stated.
The investigation was conducted rigorously meeting a high technical standard.
The methods are described with sufficient information to be reproducible.

Validity of the findings

No comments

Additional comments

* Line 122:
“There are two published systematic reviews ….”: As there are more than two, do the authors mean the two “most recent” reviews ? Please clarify ...

* Timing of probiotic initiation (Lines 343-350; 414-421):
The authors made an effort to analyze the effects of the timing of probiotic initiation: However, I am afraid it is impossible to make a meaningful conclusion from the data as they were available, unless the exact timing of probiotic initiation is known from the RCTs starting "with first feeds". "With first feeds" is overlapping very much with "right after birth", "< 48 hours", "within 72 house" or within the first week - depending on local feeding policies.

The fact that starting below 48 hours does not have an effect on any outcome is puzzling. In light of a variety of different probiotic preparations it is probably rather by chance due to the nature of the probiotic in these studies or a low number of trials rather than a true effect. What conclusion can those unit´s draw, who start their first feeds within the first 48 hours, is it of no use then ?

To my opinion it would be most honest to be discussed in the direction that no conclusion can be made unless the data on when probiotics were exactly started of those infants are available that were started on probiotics "with first feeds". As the analysis was planned a priori it cannot just be left away, but the data are rather disturbing than clarifying, so I think they need to interpreted in another way.

* Table 1: in the study by Lin et al 2005, it is B. infantis, not B bifidum.
There are two preparations of Infloran(r) being marketed by the company under the same name, one containing B. bifidum (used in the study by Lin in 2008), the other one with B. infantis (used by Lin in 2005...)

·

Basic reporting

Type of feeding (exclusive mother's milk; donor milk, preterm formula, formula, or a combination of these) should be included in the study characteristics (table 1). This is important from a clinical aspect as exclusive human milk is known to significantly reduce the incidence of NEC.

Experimental design

1. The analysis re-iterates the findings already published by Aceti et al (2015), Alfaleh et al (2014, Cochrane update), Yang et al (2014). From an epidemiological perspective, the efficacy of probiotics to reduce severe NEC and mortality has been shown with pretty good precision. If we compare the point estimates and precision (95% CIs) of all the above studies including this study under review, we do not find much of a difference in terms of the primary outcome (reduction of severe NEC). Hence, from a clinician's point of view, another meta-analysis using the classical (frequentist) approach may not add much of a value with regards to practice change. Rather, it would be methodologically more interesting if the meta-analysis was updated using Bayesian methods to see how these recent studies affect/alter already known facts.(http://handbook.cochrane.org/chapter_16/16_8_1_bayesian_methods.htm)

2. Although probiotics are proven to be beneficial, the question still remains, are they beneficial if the NEC incidence is already low? This is an argument often used by centers with relatively low NEC incidence justifying not using it. Hence from a clinical perspective it would be interesting to look at a subgroup analysis of the primary outcome(s), subgrouped according to the incidence of NEC. If probiotics are shown to be equally effective across centers with different NEC incidence rates then that would be an extremely important evidence for practice change for clinicians still reluctant to use it.

Validity of the findings

The results seem to have good internal validity. Attention to the above mentioned "experimental design" aspects should add to the generalizability of the study and make it clinically more relevant.

Additional comments

Well conducted systematic review and meta-analysis on a very important topic. However the importance of another updated meta-analysis using the traditional approach in the wake of so many others is questionable. Therefore, a Bayesian approach to the analysis may make this paper stand out both from a methodological as well as a clinical perspective. Also, incidence based subgroup analysis would also make it an interesting read for clinicians and epidemiologists alike.

---

## Round 0.2 · Major Revisions

· Academic Editor

Major Revisions

You have chosen a very unfortunate way to update the literature search. Your "systematic" review now includes studies identified by

i) previous SRs
ii) your own search from 2013 to Jan 2015
iii) a recent search of trials that were ongoing in 2013/2014

This is no longer a systematic review because there is nothing systematic about the search strategy anymore. I would ask you to either revert to your original search results or (my preference) properly update your search using your original search strategy. I am a bit surprised that re-applying an existing search strategy that yielded < 200 hits to a shorter time period would pose a problem.

---

## Round 0.3 · accepted · Accept

· Academic Editor

Accept

The authors have updated and revised their search strategy as requested.